# Studying biomolecular folding and binding using temperature-jump mass spectrometry

Adrien Marchand[1], Martin F. Czar[1], Elija N. Eggel[1], Jérôme Kaeslin[1] & Renato Zenobi [1]*

Characterizing folding and complex formation of biomolecules provides a view into their thermodynamics, kinetics and folding pathways. Deciphering kinetic intermediates is particularly important because they can often be targeted by drugs. The key advantage of native mass spectrometry over conventional methods that monitor a single observable is its ability to identify and quantify coexisting species. Here, we show the design of a temperature-jump electrospray source for mass spectrometry that allows one to perform fast kinetics experiments (0.16–32 s) at different temperatures (10–90 °C). The setup allows recording of both folding and unfolding kinetics by using temperature jumps from high to low, and low to high, temperatures. Six biological systems, ranging from peptides to proteins to DNA complexes, exemplify the use of this device. Using temperature-dependent experiments, the folding and unfolding of a DNA triplex are studied, providing detailed information on its thermodynamics and kinetics.

[1] Department of Chemistry and Applied Biosciences, ETH Zurich, CH-8093 Zurich, Switzerland. *email: zenobi@org.chem.ethz.ch

The folding energy landscape of biomolecules or biomolecular complexes is often regarded as a funnel[1,2]. In this picture, folded structures reside at the base of the funnel, and the route from the outside of the funnel to its base corresponds to the folding pathway. On the way to the final state, a biomolecule can adopt metastable conformations, or intermediates, which correspond to partially folded or misfolded structures. These local minima in the folding landscape can retain a portion of the molecular population for a while. The height of the energy barriers that this portion of population has to pass through defines how long it remains trapped[3].

Studying energy landscapes of biomolecules and biomolecular complexes is not straightforward. In chemical reactions involving small molecules, the different species are structurally and energetically well-defined because of the formation of high-energy covalent bonds; in contrast, the folding of a large biomolecule is driven by multiple noncovalent interactions. In this case, one observable state may be an ensemble of several microstates sharing similar thermodynamic and kinetic properties[4]. The picture can be even more complicated when binding partners are involved. Each of the coexisting unbound states could interact with one or multiple partners, and through one or more binding sites.

Obtaining information on the energetics of intermediates and transition states for biomolecular complexes leads to a better understanding of their stabilities, and dynamics, and could be interpreted in terms of structures[5]. For instance, when two binding partners compete for the same binding site, the relative free energy levels of the two bound states dictate the relative populations at equilibrium. Ligand-induced allosteric effects function similarly. In addition, characterizing folding intermediates is also of interest for drug design. To deactivate a biomolecule, a proportion of the population can be trapped in an inactive (e.g. misfolded) state by using specific ligands[6,7].

To characterize energetics, isothermal titration experiments allows one to obtain the equilibrium constant for the formation of multimolecular complexes[8], and thermal denaturation experiments are used to get information about the change of the equilibrium constants with temperature[9]. Decomposing the Gibbs free energy into its enthalpic and entropic contributions gives a deeper insight into the driving forces of the formation of complexes. This is of particular interest when high-resolution structures cannot be obtained[10]. Kinetics experiments give access to the mechanism and allow the study of transition states, intermediates, and kinetic traps. In a kinetics experiment, the binding or folding is initiated by changing a property of the system, pushing it out of equilibrium; the system then evolves towards its new equilibrium state.

A key element of a kinetics experiment is the trigger used to initiate the reaction. The most common trigger is the addition of a ligand (i.e., a concentration jump). To study protein folding, denaturants can be added to the solution as triggers to the unfolding kinetics, whereas re-dilution into the folding buffer allows one to monitor folding kinetics. Temperature (or, more rarely, pressure) jumps can also be used to quickly perturb equilibrium. For example, the refolding of cold-denatured proteins can be initiated by a rapid temperature increase, using laser pulses[11–13]. For technical reasons, temperature jump down approaches are more difficult to implement[14].

When multiple binding partners are involved, the simultaneous characterization of all the coexisting states is of prime importance. This is also important for intra-biomolecular folding processes because there is increasing evidence showing that many of these processes do not follow simple two-state models[15,16]. Being able to distinguish multiple states could provide insights into the binding or folding mechanism. While it is sometimes possible to decompose the signals coming from different species, classical biophysical methods usually struggle when more than two species coexist in solution, and getting a clear picture of the solution reactions can become difficult. This is because these techniques typically monitor a change in a single property of the system, irrespective of whether this change has been initiated by a change in ligand concentration or temperature[17–21].

Mass spectrometry (MS) has become a useful biophysical tool for the characterization of more complex systems[22–25]. Its main strength lies in its ability to separate species in a mixture (as long as they have different masses). Moreover, using native electrospray ionization (ESI), noncovalent complexes can be maintained from the solution to the gas phase and be distinguished based on their masses. MS becomes very useful for complexes composed of more than two partners or when multiple binding sites can be populated at the same time[26]. Time-resolved experiments are also possible[27,28]. For example, in a recent study using direct infusion, the stepwise cis–trans isomerization of polyproline peptides was tracked by native MS[29]. In order to reach shorter time scales, mixing devices have also been developed over the years[30–32]. In addition, it has even become possible to separate complexes of the same mass but of different structures (for example, conformers) by combining mass spectrometry with ion mobility[33].

Recent instrumental developments from our group and others have allowed the monitoring of the relative free energy levels of several coexisting species as a function of temperature[34–37]. For example, the Clemmer group showed that the thermal denaturation of ubiquitin does not follow a simple two-state model[35]. Different intermediate species are populated at different temperatures. In another study, we were able to monitor the evolution of the concentrations of up to eight coexisting biomolecular complexes made of a DNA strand, cations and ligands as a function of temperature[10]. Using a thermostated capillary right after a mixing tee, the Stillman group monitored the sequential binding of multiple arsenic cations to the metallothionein protein as a function of time and temperature[38]. However, while these studies provide an unprecedented level of detail of binding or folding processes, only experiments at equilibrium or after relatively long mixing times can be conducted with such approaches, because short time scales cannot be accessed (shortest ≈several minutes). It would thus be of great interest to study many relevant biological systems for which the binding/folding occurs in seconds or faster.

Here, we present a temperature-jump ESI-MS source suitable for monitoring fast kinetics at different temperatures. The ability to perform temperature jumps from both high to low and low to high temperatures allows one to study thermal unfolding as well as refolding of thermally denatured biomolecular complexes. Studies of biomolecular systems – ranging in complexity from protein folding to ternary noncovalent biomolecular complex formation and dissociation – are used to show that our method can be used to give much more information on the thermodynamics and kinetics of biomolecular folding and noncovalent binding than previously possible using MS-based methods. Six biological systems are chosen as examples: the formation of a DNA duplex, of a DNA triplex and of a DNA G-quadruplex, ligands binding to a protein, the folding of a protein, and the dissociation of a triple helix formed from collagen model peptides. To fully exploit the capabilities of the source, we record the kinetics of formation of a DNA triplex at multiple temperatures and map key parts of the energy surface for the formation of this complex.

## Results

**Design and principle of temperature-jump ESI-MS.** The source combines both flow-tube type and temperature-jump approaches.

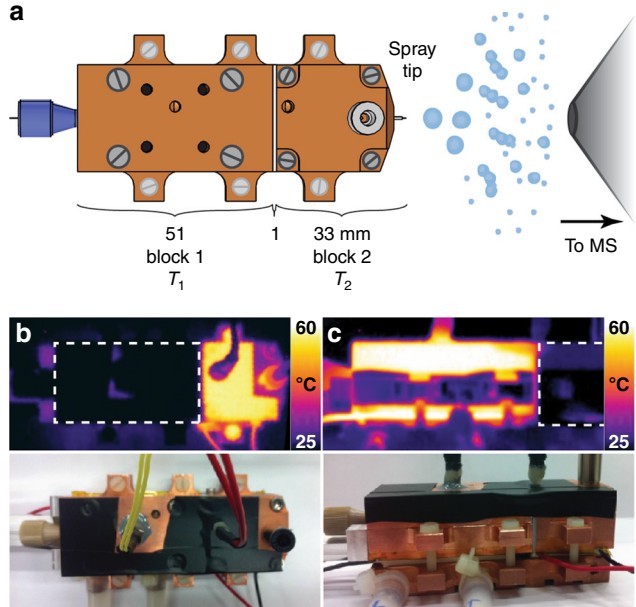

**Fig. 1 Experimental setup. a** Schematic representation of the temperature-jump electrospray source. **b**, **c** Infrared pictures of the source with blocks 1 and 2 maintained at 60 and 25 °C and the reverse. The bottom pictures show the source in similar positions. To obtain the IR pictures, black tape was placed on the source in order to enhance blackbody emission and to reduce the reflection of light coming from other sources.

In a typical experiment, a solution containing a biomolecular complex of interest is flown through a capillary embedded inside two independently temperature-controlled copper blocks (Fig. 1a and Supplementary Fig. 1). First, equilibrium at the temperature of the first block is reached. This equilibrium is then perturbed by a sudden change in temperature when the solution enters the second block. At the end of the second block, the solution is sprayed as with any pneumatically assisted ESI source and the solution is analyzed by native MS. By varying the flow rates, different residence times can be accessed and kinetics can be recorded. Using a 100-μm ID capillary and liquid flow rates between 100 and 0.5 μL/min, the time spent in the second block can be varied between 160 ms and 32 s, respectively, providing more than two orders of magnitude in the time domain. Supplementary Table 1 lists the flow rates used and the respective times reached. Recording one kinetics experiment (12–15 time points) at one temperature takes ~20–25 min and consumes around 150 μL of solution (concentration 5–40 μM). Further technical details are provided in the Materials and Methods section and in the Supplementary Fig. 1.

As confirmed using an infrared camera (Fig. 1b, c), due to the high thermal conductivity of copper, the temperatures of the two blocks were very homogeneous. The heat insulation of the 1-mm Teflon layer is sufficient to prevent the formation of a continuous temperature gradient along the source and allows an abrupt change of the temperature along the blocks. Using Comsol Multiphysics simulations, we estimate the time needed by the solution to reach the temperature of the second block and the error on the time domain for each flow rate. Supplementary Figure 2 shows the temperature of the center of the solution along the capillary from the end of the emitter. At higher flow rates, the errors relative to the residence time are larger, although small in absolute values (for flow rates below 10 μL/min, the error is maximum, at 10%; see Supplementary Table 1). Overall, because these computed error bars in the time dimension are rather small, and so as to not clutter the figures, they are not shown on any of

the plots. We note that the time needed for the desolvation process is negligible (≈μs) when compared to the time scales accessible using the $T$-jump source[39].

**Monitoring kinetics after a temperature jump**. As a model binary system, we first examined the kinetics of DNA G-quadruplex formation in the presence of potassium. Their equilibria have been widely studied by many biophysical techniques[40–43], including native ESI[10,44]. The sequence studied here is 22CTA (d(A(GGGCTA)₃GGG, PDB ID: 2KM3[45], Fig. 2a), which forms a 1-K⁺ antiparallel structure at 25 °C in the presence of potassium as indicated by the mass spectrum in Fig. 2b. We monitored the concentrations of the species at equilibrium as a function of temperature using the temperature-controlled ESI source described previously (i.e. without $T$-jump)[10]. Briefly, using this source, the whole solution is maintained at a desired temperature. As the temperature is ramped up, the complex unfolds, as indicated by the release of the bound K⁺ cation. The peaks corresponding to the 0 and 1-K⁺ stoichiometries were integrated as a function of temperature and were normalized under the assumption of equal detection efficiency[46] to produce the thermal denaturation curve shown in Fig. 2c. The measured thermal denaturation temperature ($T_m$), defined as the temperature at which half the complexes are formed, is 36 ± 1 °C (mean ± standard deviation), in good agreement with the previously reported value[10].

Using the $T$-jump source, we monitored the reincorporation of the potassium as a function of time (Fig. 2d, e). To do so, we performed a temperature jump from 75 °C, a temperature at which the complex is fully dissociated, to 25 °C, at which the G-quadruplex is the dominant species. At very high flow rates, which correspond to short time scales, the mass spectrum resembles that obtained at equilibrium at 75 °C. As the flow rate is decreased and the reaction time increases, the species containing one potassium ion become more prominent. In this case, at 25 °C, the equilibrium is reached after ~15 s. At a reaction time of 32.6 s, the mass spectrum is similar to that obtained at equilibrium at 25 °C. We note that in another study using classical infusion it was not possible to record the kinetics of binding of K⁺ to this sequence because it happened within the dead time of the experiment[44]. The solid curves in Fig. 2e are obtained by fitting the rate constants from kinetic equations for the chemical reaction described in Fig. 2a (see Methods for more details). The fitted association and dissociation rate constants are given as insert in Fig. 2e. Folding kinetics for the same system were monitored at different final temperatures: as the temperature increases, the reaction rates become larger (Supplementary Fig. 3).

We note that for this system, equilibrium is established in the first block within the residence time in the first block. This was checked by setting the temperature of the first block to room temperature and the second block to 75 °C. At this temperature, even at the highest flow rates, the observed signals correspond to the DNA without any cation bound, indicating that complete denaturation is achieved at 75 °C in <160 ms (Supplementary Fig. 4). Additionally, we recorded the ion mobility of the 5- and 6-charge states as a function of temperature and time (Supplementary Fig. 5). The mobility of the 0-K⁺ stoichiometry does not change, indicating that the release of K⁺ is concomitant with the unfolding of the G-quadruplex. We also note that no other products were produced after maintaining the DNA at high temperature for longer times (Supplementary Fig. 6).

We also checked that the flow rate itself does not influence the distributions of the detected species. As shown in Supplementary Fig. 7, except for the appearance of a trimethylammonium adduct

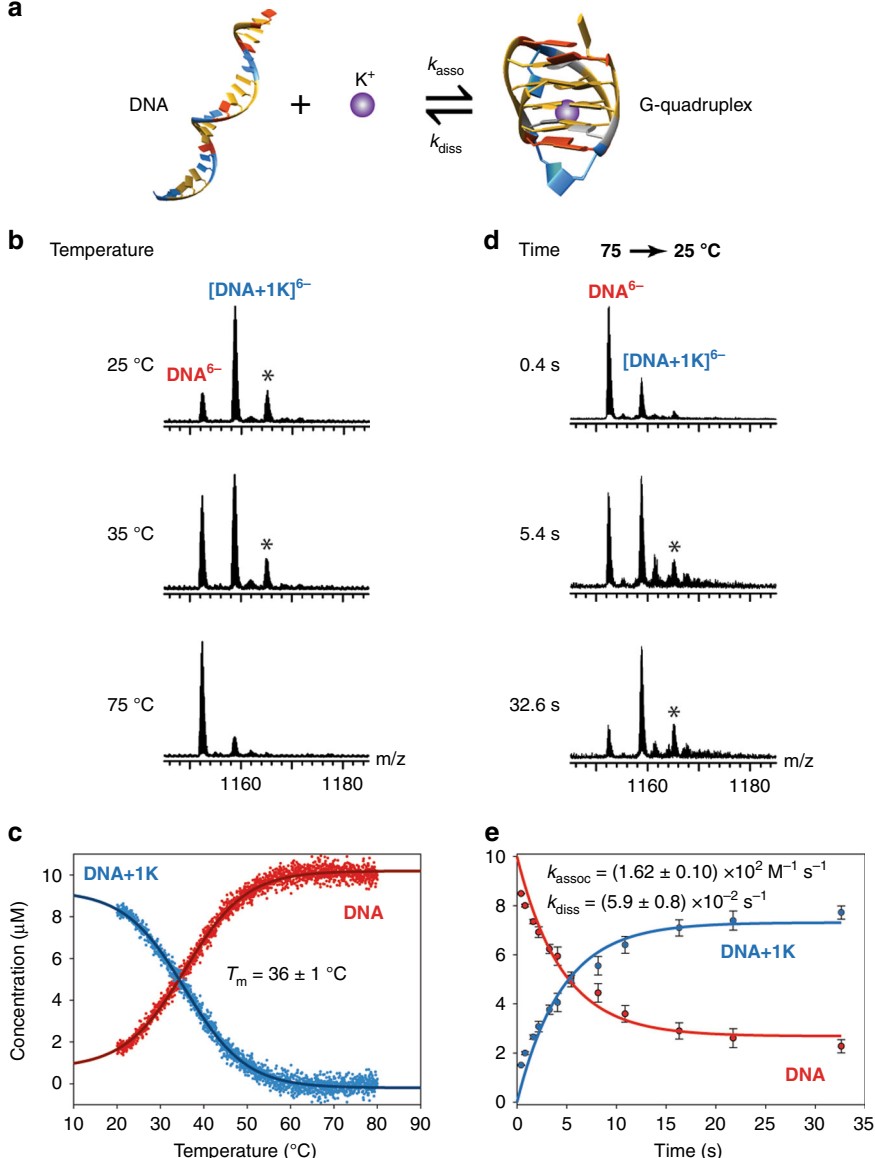

**Fig. 2 Kinetics of G-quadruplex formation. a** Representation of the chemical equilibrium of binding of K$^+$ to the DNA sequence 22CTA (d(A (GGGCTA)$_3$GGG), PDB ID: 2KM3[45]). **b**, **c** Thermal denaturation of 22CTA–K$^+$, monitored by MS (10 μM DNA in 100 mM TMAA and 1 mM KCl). **d**, **e** Kinetics recorded using the *T*-jump ESI source (jump from 75 to 25 °C). The mass spectra are magnified views of the 6- charge state. The peaks labeled with a star correspond to a nonspecifically bound K$^+$ ion. **c**, **e** Quantification of the species taking into account nonspecific adducts as described previously[47,48]. The 5- and 6- charge states were averaged. The vertical error bars in **e** are the standard deviation obtained from the quantification on different charge states (5- and 6-, *n* = 2). The reproducibility of the experiment is discussed in the Discussion section. The errors on the rate constants are the standard deviations from the fitting. Source data are provided as a Source Data file.

from the buffer at low flow rates, the spectra are identical. For every system discussed in this manuscript, we systematically performed these controls before recording the kinetics. If equilibrium was not reached, the time point was discarded.

**General applicability of the temperature-jump source**. We next established the applicability of the source over a wide range of temperatures, rate constants and biological systems. First, as an example of dimerization, we studied the formation of a DNA duplex formed by a self-complementary sequence (d(CGT AAA TTT ACG; sequence adapted from reference[49]) (Fig. 3a–c and Supplementary Fig. 8). At equilibrium (at 25 °C), the duplex is formed (Supplementary Fig. 8). At higher temperatures, the duplex dissociates into monomers; the thermal denaturation

temperature is 53 ± 2 °C. The kinetics of refolding of the duplex were studied by performing a temperature jump from 75 to 25 °C (Fig. 3b). On short time scales, mostly monomer peaks are present. They disappear with time to form the duplex. Note that the monomers and dimers overlap for some charge states; these were separated using ion mobility spectrometry, based on differences in charge, mass, and shape[33].

Secondly, to show that the source is not limited to study DNA complexes and can be used to monitor the kinetics of protein-ligand interaction, we studied the binding of the protein carbonic anhydrase II (PDB ID: 1CA2[50]) to two ligands: 4-carboxybenzene sulfonamide and benzene sulfonamide (Fig. 3d and Supplementary Fig. 9). At room temperature, the ligands are mostly bound and are released at higher temperatures to form the free protein and an adduct of the protein with an acetic acid molecule (Supplementary

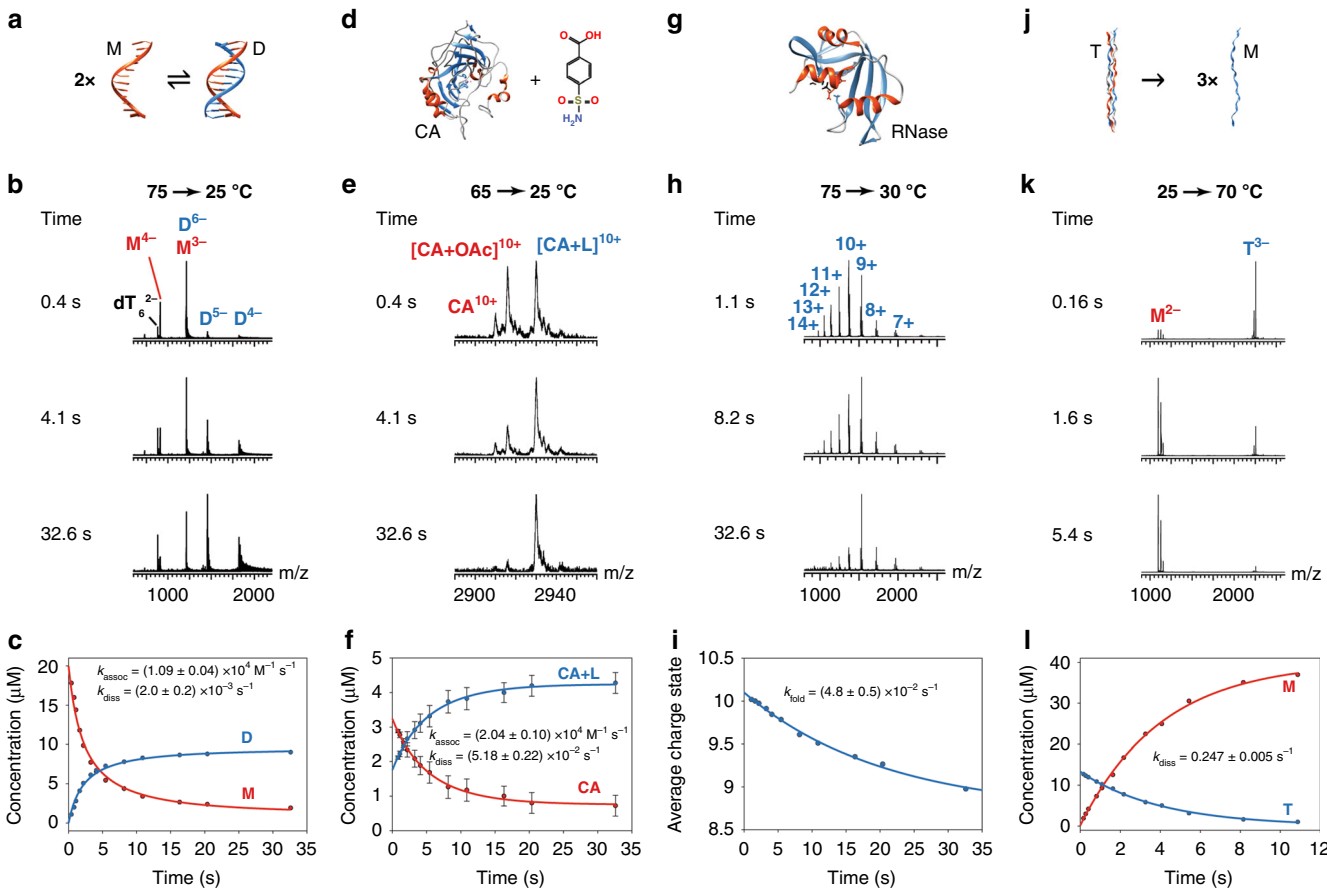

**Fig. 3 Kinetics recorded using the temperature-jump source for exemplary biological systems. a–c** the formation of a self-complementary DNA duplex (20 μM d(CGT AAA TTT ACG) in 100 mM TMAA). M and D denote the monomer and duplex, respectively; **d–f** the binding of a ligand to a protein (5 μM Carbonic Anhydrase II (PDB ID: 1CA2[50]) with 5 μM 4-carboxybenzene sulfonamide in 10 mM NH₄OAc). CA and L denote Carbonic Anhydrase II and ligand, respectively; **g–i** the folding of a protein (10 μM Ribonuclease A (PDB ID: 5RSA[80]) in 100 mM NH₄OAc at pH 2.75). RNase denotes Ribonuclease A; **j–l** the dissociation of a collagen peptide triple helix into monomers (40 μM [POG]₈ collagen peptide (PDB ID: 1CGD[54]) in 10 mM NH₄OAc). M and T denote monomer and trimer, respectively. The vertical error bars in **f** are the standard deviation obtained from the quantification on different charge states (12+, 11+, 10+, and 9+, $n = 4$). The reproducibility of the experiment is discussed in the Discussion section. The errors on the rate constants are the standard deviations from the fitting. Source data are provided as a Source Data file.

Fig. 9). Both were considered as free from ligand for the quantification. The temperature at which 50% of the protein binds a ligand is $51 \pm 2\,°C$ and $39 \pm 4\,°C$ for 4-carboxybenzene sulfonamide and benzene sulfonamide, respectively. We note that the thermal denaturation transition from bound to unbound ligand occurs over a very wide temperature range, suggesting low values for the entropies of binding, as previously reported for similar protein-ligand systems[36,51]. However, the amount of complex changes enough with temperature so that a jump from 65 °C to 25 °C reveals the kinetics of binding. At a refolding temperature of 25 °C, almost half the complex is re-associated within 0.4 s, and equilibrium is reached within 30 s (Fig. 3d–f and Supplementary Fig. 9). Interestingly, the association rate constants for the ligands at 25 °C differ by a factor of four, whereas the dissociation rate constants are similar. Thus, the difference in the association rate constants for these two ligands explains the difference in their relative binding affinities, which is information that is not attainable by equilibrium measurements alone.

Thirdly, to show that the source can also be used to study the kinetics of protein folding by charge state distribution monitoring[52], we examined the refolding of the small protein Ribonuclease A after thermal denaturation (Fig. 3g–i and Supplementary Fig. 10). At equilibrium, at low temperature, the charge state distribution is narrow and centered on the 9+ charge state. When

the solution is heated, the charge state distribution shifts towards higher values (11+), indicating that the protein is unfolding (Supplementary Fig. 10). The measured thermal denaturation temperature is $46.1 \pm 0.2\,°C$. A temperature jump from 75 to 30 °C allows monitoring the refolding of the protein. Interestingly, at very short time scales, the average charge state is 10+, suggesting that the first step(s) of the refolding of the protein (during which the average charge state changes from 11+ to 10+) occurs in the dead time of the experiment (in this case, 160 ms). The second part of the kinetics is much slower and well fitted by a single exponential function. The equilibrium is reached after more than 30 s. This multi-step refolding of Ribonuclease A was previously described using UV absorption temperature-jump experiments[53] and the last step, also studied by NMR, was reported to be slow[13].

Finally, the source can also be used to monitor the unfolding of complexes by jumping from low to high temperature. To show this, we used a collagen model peptide ([Proline-Hydroxyproline-Glycine]₈, [POG]₈; PDB ID: 1CGD[54]). Such peptides form triple helical structures and are used as model peptides to study the assembly of collagen[55,56]. Their folding and unfolding behavior has been reported to be extremely slow, on the order of days at room temperature[57]. Here, we show that their unfolding is strongly temperature dependent (Supplementary Fig. 11), with rate constants for dissociation being ten times higher for every

10 °C temperature difference; the dissociation rate constant increases 200-fold when changing dissociation temperature from 50 to 73 °C. For example, we show in Fig. 3k-l that the equilibrium is reached within 12 s at 70 °C, whereas at 60 °C it takes more than 3 min.

**Building a biomolecular energy surface**. In the following section, we show how $T$-jump ESI-MS can be used to survey parts of the energy landscape of a biomolecular complex. To do so, we recorded equilibria and kinetics at different temperatures. Van't

Hoff analysis and transition state theory were used to obtain the relative energy levels of the different species. We chose a DNA triplex as a model system because such systems have been widely studied[58,59], and they have been implicated in gene expression and human diseases[60].

In a triplex, Watson-Crick base pairs are formed between two antiparallel strands that form a duplex, and a third strand binds in the major groove, forming Hoogsteen-type hydrogen bonds[61]. The exact sequence we chose forms a triplex whose structure has been solved by NMR (Fig. 4a, PDB ID: 1BWG[62]). In this complex, the Hoogsteen strand is parallel with respect to the

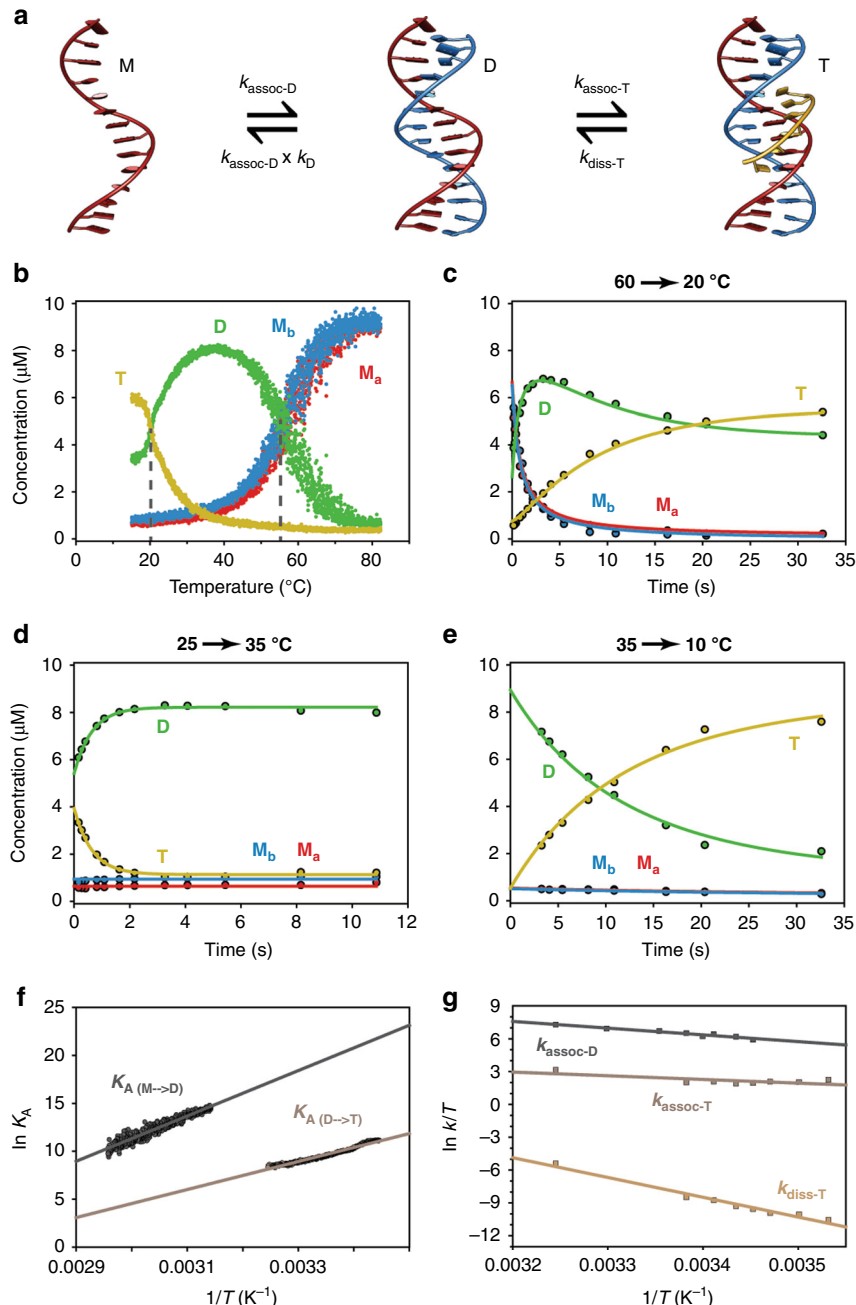

**Fig. 4 Thermal denaturation and kinetics of folding and unfolding of a DNA triplex. a** Representation of the chemical equilibrium of formation of a DNA triplex. **b** Thermal denaturation experiment of a DNA triplex monitored by native MS (10 μM $M_a$, 10 μM $M_b$, and 30 μM $M_c$ in 100 mM TMAA at pH 5.5). **c-e** Kinetics of triplex formation monitored using $T$-jump ESI-MS, using jumps from from 60 to 20 °C, 25-35 °C, and 35-10 °C, as indicated. For the kinetics, a model corresponding to the chemical equations displayed in **a** was fitted to the data to obtain the rate constants. **f** Van't Hoff plots for the formation of the duplex and the triplex. The intercept is proportional to $\Delta S^0$ and the slope to $\Delta H^0$. **g** Eyring plot for the formation of the duplex and the formation and dissociation of the triplex. The intercept is proportional to $\Delta S^\ddagger$ and the slope to $\Delta H^\ddagger$.

strand of the duplex to which it is bound, and two types of base triplets are formed: TA•T triplets and CG•C$^+$ triplets where the cytosine bases of the Hoogsteen strand are required to be protonated to bind to the duplex. To allow formation of the triplex, a pH of 5.5 was used. Therefore, multiple stoichiometric species (monomer, duplex and triplex), which are all easily identified by MS, are expected to be simultaneously present in solution, depending on temperature, reaction time, and concentrations.

First, we monitored the evolution of equilibrium concentrations of the different species as a function of temperature (Fig. 4b). At low temperatures, mostly triplex is detected. This species disappears when the temperature increases, and has a melting point of $20 \pm 2\,°C$. At 35–45 °C, the duplex is predominant. The duplex is denatured at $55 \pm 1\,°C$. Both thermal denaturation temperatures are very similar to those obtained by CD when monitoring specific wavelengths as previously demonstrated for other DNA complexes[10] and for collagen triple helices[63] (21 °C at 269 nm and 53 °C at 242 nm for the triplex–duplex and duplex–monomer transitions, respectively; Supplementary Fig. 12). When monitoring other wavelengths such as 281 nm, the thermal denaturation curves obtained by CD are not sigmoidal and it is clear that more than two states coexist in solution. These results confirm that MS is a suitable technique to study DNA triplexes, as also previously reported with room temperature experiments[64,65].

We then performed temperature-jump experiments to monitor the simultaneous formation of the two complexes and their interconversion. In the first set of experiments, the temperature of the first block was set to 60 °C, above the determined melting temperatures of both complexes. At 60 °C, the main species in solution are the monomers with a small fraction (≈30%) remaining in the form of duplex. The temperature of the second block was set to 20 °C to allow refolding of both duplex and triplex. As shown in Fig. 4c, at short time scales, mostly monomers are detected and, as more time is allowed for the reaction to occur, duplex and triplex species are formed. As expected for such complexes, we found that the duplex is formed first, within a few seconds. Triplex formation is slower and the equilibrium is not yet reached after 32 s at 20 °C. We also performed jumps from 60 °C to different temperatures between 16.5 and 22.5 °C (Supplementary Fig. 13) and ran experiments without the third strand, to monitor the formation of the duplex only (Supplementary Fig. 14).

As previously demonstrated with the collagen triple helix, the source also allows jumps from low to high temperatures. This allowed examination of the unfolding of these complexes, as shown in Fig. 4d, with a temperature-jump from 25 to 35 °C. In this case, a temperature of 25 °C was chosen for the first block as it is also the temperature of the laboratory; we were therefore assured to be at equilibrium. Within two seconds, the new equilibrium at 35 °C is reached and almost only duplexes remain in solution. Finally, to record the kinetics at very low temperatures, we set the temperature of the first block to 35 °C and performed jumps to 10, 12.5, and 15 °C (Fig. 4e and Supplementary Fig. 15). Under these conditions, mostly duplexes are already present in solution and the formation of the triplex is the only reaction expected. This is exactly what was observed if the time points below 2 s, for which the equilibrium is not yet reached in the first block (see Fig. 4d), are excluded from the analysis. This is a typical example showing that it is necessary to check that the equilibrium is reached in the first block by inverting the temperatures of the two blocks.

We used the same kinetic model, displayed in Fig. 4a, for each experimental dataset that we recorded and the fits are shown in Fig. 4c–e and Supplementary Figs. 13–15. This model involves

sequential formation of the duplex and the triplex. The fit was found to be excellent in every case. We simplified the model by including the equilibrium constant calculated using the temperature-controlled nanospray source (Fig. 4b) because, at equilibrium at low temperature, the monomer concentrations are very low and would entail large errors in the obtained rate constants. This allowed reducing the number of fitted parameters while giving reliable kinetic constants. For each temperature, we obtained a combination of rate constants, the values for which are given in the Supplementary Table 2. The folding of the complexes proceeds more rapidly with increasing temperature, as expected for Arrhenius-type kinetics. Interestingly, overall, the kinetics of association of the triplex is almost temperature independent.

We then used the equilibrium concentrations obtained as a function of temperature from the thermal denaturation experiments to decompose the Gibbs free energy into the enthalpies and entropies of formation of the duplex and the triplex, using a Van't Hoff plot (Eq. 1, Fig. 4f). Similarly, we applied the transition state theory to decipher the folding of this complex. A linear fit of the Eyring plot (Eq. 2, Fig. 4g) gives access to the enthalpies and entropies of activation. From the enthalpies and entropies of formation, we obtained three Gibbs free energies for the formation of the transition state complexes. The values obtained are shown in Fig. 5. We note that the validity of these analyses relies on the assumption that the enthalpies and entropies are only very weakly dependent on temperature.

## Discussion

By combining the information on the transition states energies and the energy levels obtained for each species, we were able to obtain key information about the energy landscape for the formation of this multimolecular complex. In Fig. 5a, we show the Gibbs free energy at 298 K for the formation of the duplex and triplex as well as their transition states. The Gibbs free energy drops sequentially as the duplex and the triplex are formed, and the two barriers are similar. To obtain deeper insight into the folding mechanism, we considered at the enthalpic and entropic contributions for each sequential step in the folding mechanism.

When duplexes and triplexes are formed, hydrogen bonds are created, leading to negative enthalpies (Fig. 5b). The transition state enthalpies are, however, positive, with the transition state enthalpy of association of the triplex being smaller than that for the formation of the duplex ($\Delta H^{\ddagger}_{assoc-T} < \Delta H^{\ddagger}_{assoc-D}$); this is expected because fewer new hydrogen bonds are created to form a triplex from a duplex, than to form a duplex from monomers. The activation energy ($E_A$) obtained from Arrhenius theory is related to the transition state enthalpy ($\Delta H^{\ddagger}$), for bimolecular complexes, according to the following equation:

$$E_A = \Delta H^{\ddagger} + R \cdot T \qquad (1)$$

Where $R$ is the ideal gas constant and $T$ the temperature. Our value for the formation of triplexes agrees with the small (or even negative) values of activation energies previously reported for similar complexes[66].

Secondly, both the entropy of formation of the duplex and the triplex are unfavorable, as expected for the formation of multimolecular complexes. Interestingly, the transition state entropies for the association of the duplex and the triplex have different signs, suggesting different mechanisms. With a positive $\Delta S^{\ddagger}$, the formation of the duplex is thought to proceed through a dissociative mechanism[67,68]. This means that the DNA bases first have to be freed from interacting groups (i.e., solvent molecules) before the duplex can be formed. In contrast, the formation of the triplex presents a negative $\Delta S^{\ddagger}$, which presumably corresponds to an associative mechanism, in which, similarly to organic

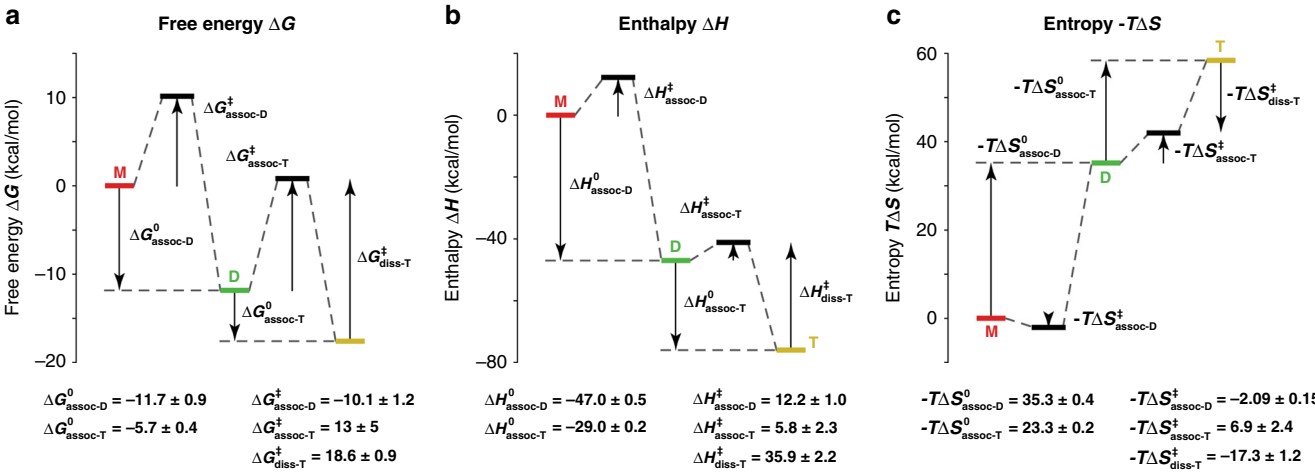

**Fig. 5 Energy surfaces obtained from the combined equilibria and kinetics experiments.** The experiments were monitored using mass spectrometry for the formation of the DNA duplex and triplex in 100 mM TMAA at pH 5.5: **a** Gibbs free energies, $\Delta G$ at 298 K, **b** Enthalpies, $\Delta H$, and **c** Entropies, $-T.\Delta S$ at 298 K. The y-axis corresponds to the calculated energies and the x-axis to the reaction coordinate. The values are reported in kcal/mol. The errors are the standard deviation obtained from the Van't Hoff or Eyring analyses of Fig. 4.

chemistry, the nucleophilic attack precedes the departure of the leaving group ($S_N2$ type reaction). We interpret this negative value mechanistically by the concomitant desolvation of the heavily solvated major grooves of DNA duplexes[69] and the binding of the third strand to form the triplex. The mechanism of formation of triplexes have previously been debated[66] and is assumed to proceed through a zipping mechanism, which would be in agreement with the thermodynamic parameters calculated here.

For the purpose of validation, we compared the values we obtained with literature data whenever possible. In this case, SPR experiments were previously performed to determine the rate constants for the binding of the ligands we used in this study to Carbonic Anhydrase II[70]. Also, the folding of Ribonuclease A was studied using NMR temperature jumps[13]. The values are in very good agreement for both systems (See Supplementary Table 3 for a comparison of the values)[70].

Often, however, kinetics data were not available. In these cases, we compared the enthalpies and entropies of formation of the complexes obtained from the thermal denaturation experiments (from Van't Hoff plots) to those obtained from the kinetics experiments (by subtracting the association and dissociation activation enthalpies and entropies, see Fig. 5). The comparisons are shown for the formation of the triplex, the DK33 duplex and the 22CTA G-quadruplex respectively in Supplementary Figs. 16–18. The values calculated using the two methods agree within 10–20% and are also consistent with the values calculated using Van't Hoff plots from CD experiments.

We evaluated the reproducibility of the experiments by performing the same temperature jump for the formation of the duplex (Supplementary Fig. 19) at 35 °C. Even for such fast kinetics (equilibrium reached in <5 s), the reproducibility is excellent and the values for the association rate constant has a relative standard deviation of 4–18%, typical for mass spectrometry quantification[71]. In addition, to prove that the temperature from which the jump is initiated does not influence the obtained kinetic constant, we compared the values obtained by starting from three different temperatures. This strongly suggests that any lag phase due to the change in temperature of the solution inside the capillary is negligible. Overall, a typical relative standard deviation of ~10% was found in the obtained rate constants from reproduced experiments.

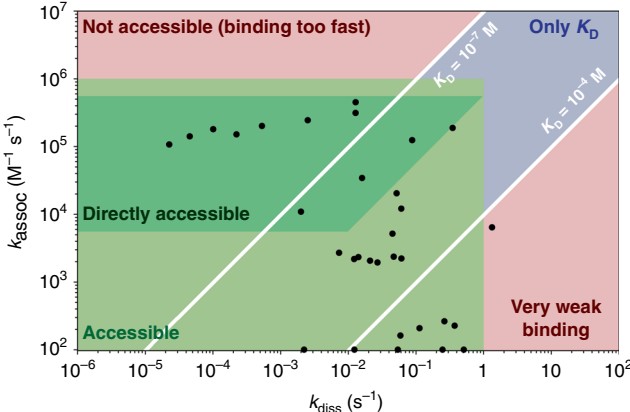

**Fig. 6 $k_{assoc}$ and $k_{diss}$ for bimolecular interactions compatible with the temperature-jump source.** The compatible area, represented in green, was plotted based on three strict criteria: (1) the time needed to fold/unfold half the complexes is longer than 0.16 s ($t_{1/2} > 160$ ms); (2) the plateau is reached in the time scale of the experiment (32 s); and (3) the concentration of the folded species changes by at least 20% of the total concentration. In dark green, the kinetics can be recorded directly at 10 μM in both target and ligand. Light green: the concentrations of the ligand and/or the target have to be adjusted to be able to access kinetics (concentrations decreased to 1 μM in both ligand and target if the reaction is too fast and concentration of the ligand increased to 100 μM or more if the reaction is too slow). Blue: the kinetics are too fast for the source and only the equilibrium constant is accessible with good confidence using mass spectrometry. Red: combinations that are not compatible with the current setup. Note that on the scheme the zones are defined with hard borders; however, in practice, the compatible zone is not so strictly defined. The black dots correspond to the values determined for the multimolecular biological systems reported in this manuscript. Since the fittings for the unfolding of the collagen model peptides were performed only considering the unfolding reactions, the dots were placed on the x-axis.

We also compared our results with data that can be obtained by other techniques. In Fig. 6, we represent the ranges of rate constants that are measurable for a bimolecular reaction with the current design of the source (Green area). With the current setup, the shortest time point is 160 ms, far above the timescales needed

for fast folding that can be accessed with faster temperature-jump techniques such as laser temperature jumps, which are able to record kinetics on the microsecond time scale[72,73]. However, the accessible time range is very convenient to study multimolecular complex formation, which is also a strength of native mass spectrometry. We also showed that slow folding can easily be monitored. The range of rate constants that are accessible is very similar to those available with the reference technique for kinetics, namely surface plasmon resonance (SPR), which records on the timescale of minutes at nM concentrations, but requires immobilization of one of the binding partners to a surface[74]. The range of rate constants we can access is also broader than the range attainable by isothermal calorimetry (ITC), which requires minutes at ≈100 μM, limiting its use to slow kinetics ($k_{assoc} \lesssim 10^4$ $M^{-1} s^{-1}$)[21,75]. Stopped-flow approaches have also been shown to give access to timescales of a few hundreds of milliseconds[11,40], which is very similar to what can be achieved with our source. The main advantage of our source over other devices that can be coupled to MS instruments (e.g., mixing tees, stopped-flow devices) is its ability to record kinetics at different temperatures. In addition, using mixing devices, organic solvents are often used to induce the folding or unfolding and, even if diluted, the co-solvent will always be seen as a contaminant. The mixing itself is also sometimes not perfect. Temperature control has also been implemented for mixing approaches but only very slow kinetics (several hundreds of seconds) could be monitored[38].

To study kinetics of biomolecular assemblies, one of the challenges is to resolve different ensembles. This is the reason why temperature jump approaches coupled with NMR have also been developed[13,76,77]. However, for fast temperature changes, only devices allowing jumps from low to high temperature have been developed thus far, mainly due to technical limitations; this limits studies to those on unfolding kinetics or renaturation of cold-denatured proteins. With the source developed here, temperature jumps from high to low are enabled, which broadens the range of applications (applicable to the folding of nucleic acids for example). In addition, relatively low amounts of samples (150 μL of 10 μM concentration or $1.5 \times 10^{-9}$ moles per kinetic run) are required.

One of the limitations of the source is that the equilibrium being studied has to be temperature dependent in the temperature range that we can access. The Gibbs free energy for a given transition changes with temperature proportionally to the entropy change of the transition. The entropy can be positive or negative and the association of the complex be favored or unfavored with increasing temperature. Technically, the temperature jumps can be reversed if monitoring of the association or dissociation is needed. We have shown here that even for equilibria that are very weakly temperature-dependent (Carbonic Anhydrase II with ligands), we could monitor the binding kinetics. Therefore, this approach would not work only when the entropy is equal to zero.

In summary, we have designed, built and characterized a temperature-controlled electrospray source for mass spectrometers that allows one to record kinetics at desired temperatures and get insights into molecular recognition and folding. The source exploits the temperature-dependence of chemical equilibria to initiate reactions. Temperature jumps from high to low temperatures or from low to high temperatures were used to monitor the association and dissociation of multiple partner complexes as well as the folding of a protein. We illustrated how the source can be used to study biological systems of various nature, ranging from the formation of DNA duplexes to protein folding or protein-ligands interactions. The method is easy to apply, robust and reproducible. It requires low amounts of sample and allows the determination of kinetic rate constants as a function of temperature.

Reaction times as short as 160 ms and as long as 32 s can be monitored, which corresponds to a range of rate constants very similar to that of the reference technique, SPR, and compares favorably with other methods such as ITC. The advantages of combining such an approach with mass spectrometry are numerous. The most obvious advantage lies in the capabilities of native mass spectrometry to identify and quantify coexisting species. Using a DNA triplex as example, we showed how the folding potential energy surface of a multimolecular complex can be deciphered to provide detailed information on its thermodynamics and kinetics.

We envision that combining our method with top-down fragmentation, ion mobility spectrometry or ion-molecule reactions to structurally characterize kinetic intermediates will provide new insights into folding/binding mechanisms.

## Methods

**Materials**. The DNA sequences were purchased from Eurogentec (Seraing, Belgium) in a lyophilized form and purified by RP-cartridge. Stock solutions were prepared at 800 μM. Lyophilized CMPs were synthesized as previously reported (Submitted) and dissolved in ammonium acetate solution (10 mM, pH 7.0) at a concentration of 4 mM before being diluted to 40 μM. For the CMPs, the solutions were annealed at 65 °C for 15 min and kept at 4 °C for at least 5 days prior to the analysis. Carbonic Anhydrase II from bovine erythrocytes (Uniprot number: P00921, Sigma-Fine Chemicals) and Ribonuclease A from bovine pancreas (Uniprot number: P61823, Sigma-Fine Chemicals) were prepared at stock concentrations of 1 mM and 475 μM in water, respectively, before dilution to 5 and 10 μM. The exact concentrations of the DNA samples were measured by UV-vis absorbance at 260 nm using the extinction coefficients provided by Eurogentec. The concentration of Carbonic Anhydrase II and the [POG]8 peptide were also checked using UV absorbance at 280 and 205 nm, respectively ($\varepsilon^{280}_{CA} = 50420$ $M^{-1} cm^{-1}$; $\varepsilon^{205}_{[POG]8} = 63940$ $M^{-1} cm^{-1}$). The concentration of Ribonuclease A was determined from precise weighing. The ligands 4-Carboxybenzene sulfonamide and Carboxybenzene sulfonamide were prepared at 1 mM concentration in 99:1 $H_2O$: DMSO. Trimethylammonium acetate and ammonium acetate were purchased from Sigma (Fluka, 1 M, for HPLC) and used as buffers. The pH of 2.75 and 5.5 were adjusted from ammonium acetate and trimethylammonium acetate solutions using glacial acetic acid. The water was DEPC-treated from Fisher BioReagents (Filtered and Autoclaved. DNase, RNase and Protease Free).

**Temperature jump ESI source**. A stainless steel capillary passes through the center of two adjacent copper blocks. The two blocks, separated by a 1-mm Teflon layer, can be independently maintained at two different temperatures thanks to two Peltier elements mounted on the side. The Peltier elements are both connected to another copper block that is maintained at room temperature using a water-cooling system. Temperature sensors, embedded into electrically insulating ceramic shells, were placed as close as possible to the center of the blocks. Due to the limited power of the Peltier elements, the maximum temperature difference between the two blocks that we could reach was ~50 °C. The copper blocks, the Teflon layer, the water-cooling system, the temperature probes (Pt-100) and the temperature controller were all designed and built in house.

Stainless steel capillaries with two different inner diameters (50 and 100 μm) were tested and the results shown here were obtained with the 100 μm capillary that gave a more intense and stable signal and was less prone to clogging. The Peltier elements were purchased from RS Components GmbH (25 × 25 mm, 20.9 W, 2.2 A, 15.7 V and 40 × 23 mm, 36.3 W, 3.9 A, 16 V) and fed by a 14-V 4A output power supply (Model number TNQX010, Matsushita Electric Industrial Co. Ltd., Japan). The stainless steel capillary was purchased from MS Wil (New Objective, Metal Tapertip, custom length: 15 cm). The temperature is controlled by a LabView software to allow user-friendly operation of the device. The solutions were continuously infused using a 500-μL Hamilton syringe and the flow rates were controlled using a Chemyx Inc. Fusion 400 syringe pump also operated using a LabView code.

Due to the relatively high flow rates as compared to traditional nanoelectrospray sources, a coaxial desolvation gas ($N_2$) is required in order to generate an ESI spray and allow the desolvation of the ions. This gas is also thermalized to the temperature of the second block using an extended gas line (≈14 cm) directly embedded in the design of this block (Supplementary Fig. 1). The nitrogen flow rates used for the coaxial desolvation ranged between 200 and 220 mL min$^{-1}$.

**Mass spectrometry**. The source was mounted held in place by a xyz translation stage on a home-made holder in front of a Synapt G2S (Waters, Manchester, UK) at ≈1 cm from the MS inlet in order to maximize signal intensity. We note that there is no effect of the distance between the outlet of the capillary and the inlet of the MS on the results of the kinetics (Supplementary Fig. 20). This is expected

because the desolvation occurs on the timescale of µs[39], whereas the kinetics that we monitor are on the timescale of ms. The electrospray voltage was set to 3.5–3.8 kV using the internal power supply from the instrument. The instrument was in resolution mode with ion mobility enabled. It was tuned to maintain noncovalent complexes in the gas phase. Key parameters are as follows. The sampling cone and source offset were set to 0 V. The source temperature was kept at 30 °C. The trap and transfer collision energies were both set to 2 V. The gas flows were adjusted to 1 mL min$^{-1}$ of Ar in the trap, 200 mL/min of He and 70 mL/min of $N_2$ in the IMS cell. The instrument was calibrated using a CsI solution at 10 mg/mL in $H_2O$:ACN.

**Circular dichroism.** CD spectra were recorded using a 0.1-cm path length Quartz cuvette (Hellma Analytics) with an Applied Photophysics Chirascan plus instrument (UK). The scanning range was 240–320 nm with a step of 3 nm. Such parameters allowed the complete recording of the spectra in 27 s. Temperature ramps of 1 °C min$^{-1}$ were used. This allowed one complete spectrum to be recorded in a temperature range of 0.5 °C. A baseline obtained from pure water at 20 °C was subtracted from each recorded spectrum. The following equation was used to convert the recoded ellipticities ($\theta$, in mdeg) into molar circular-dichroic absorption ($\Delta\varepsilon$, in cm$^2$ mmol$^{-1}$) based on the DNA concentration ($C$, 10$^{-5}$ mol L$^{-1}$):

$$\Delta\varepsilon = \theta/(32980 \cdot C \cdot l) \tag{2}$$

Where $l$ is the pathlength in cm (0.1 cm).

**Data treatment specific to thermal denaturation experiments.** The signals corresponding to the different species were extracted using Driftscope (Supplementary Fig. 21). For the duplex, the triplex and the triple helical peptide, the signals of different species at different charge states were integrated to give signal intensity versus retention time data. The retention time was then transformed in temperature using a linear correlation (a ramp of 2 °C min$^{-1}$ was used). For the G-quadruplex and the Carbonic Anhydrase-ligand systems, the most intense charge states were treated separately and averaged. For every system, the integrated intensities were then normalized between 0 and the total concentration, considering equal response factors[78]. For the folding of Ribonuclease A, the average charge state was used instead as it is more common in the literature[25].

We describe here the procedure and equations used for the determination of the DNA triplex's thermodynamic data. Similar procedures with adapted equations were used for each studied system. Once the concentrations of each species is determined, the equilibrium constants at each temperature can readily be calculated using:

$$K_{D-D} = \frac{[M_A][M_B]}{[D]} \tag{3}$$

$$K_{D-T} = \frac{[D][M_C]}{[T]} \tag{4}$$

Using a Van't Hoff plot, plotting $ln K$ as a function of $1/T$, it is possible to access to the enthalpy $\Delta H^0$ and entropy $\Delta S^0$ of a transition (considering a null heat capacity):

$$\ln K = -\frac{\Delta H^0}{RT} + \frac{\Delta S^0}{R} \tag{5}$$

With $R$, the ideal gas constant (1.987 cal K$^{-1}$ mol$^{-1}$). Given thermodynamic values are reported for the association reaction.

**Data treatment specific to kinetic experiments.** Typical raw data obtained from kinetic experiments are shown in Supplementary Fig. 22 where the total ion current and the extracted signals for the monomers A and B, the dimer and triplex (all charge states summed) is represented. Because the signals are more intense at higher flow rates and to spare some sample, the recording times are reduced with increasing flow rates. In between two flow rates, the syringe pump was stopped for 6 seconds to facilitate the data treatment. To construct the kinetic plots, we performed experiments using 12–15 different flow rates. To proceed with the quantification, signals are averaged for each flow rates (not considering the first few scans of each flow rate to allow the spray to stabilize).

For each kinetics performed at each temperature, after integration and normalization considering equal response factors, the concentrations of the species is obtained at different residence times. A fit of the kinetic data obtained at each temperature was the performed using Biokin DynaFit (see commented Supplementary Note 1)[79]. By iterations, DynaFit uses the differential kinetic equations derived from a chemical model to obtain the rate constants. In the case of the DNA triplex presented, the model and equations are:

$$M_A + M_B + M_C \underset{k_{diss-D}}{\overset{k_{assoc-D}}{\rightleftharpoons}} D + M_C \underset{k_{diss-T}}{\overset{k_{assoc-T}}{\rightleftharpoons}} T \tag{6}$$

$$\frac{d[M_A]}{dt} = -k_{assoc-D}[M_A][M_B] + k_{diss-D}[D] \tag{7}$$

$$\frac{d[M_B]}{dt} = -k_{assoc-D}[M_A][M_B] + k_{diss-D}[D] \tag{8}$$

$$\frac{d[M_C]}{dt} = -k_{assoc-T}[D][M_C] + k_{diss-T}[T] \tag{9}$$

$$\frac{d[D]}{dt} = k_{assoc-D}[M_A][M_B] - k_{diss-D}[D] - k_{assoc-T}[D][M_C] + k_{diss-T}[T] \tag{10}$$

$$\frac{d[T]}{dt} = k_{assoc-T}[D][M_C] - k_{diss-T}[T] \tag{11}$$

With $M_A$ and $M_B$ the complementary strands and $M_C$, the triplex strand, D and T stand for dimer and trimer, respectively, as described in Fig. 4a. $k_{assoc-D}$, $k_{diss-D}$, $k_{assoc-T}$, and $k_{diss-T}$ are the rate constants for the association and the dissociation of the dimer and the association and dissociation of the trimer, respectively, as described in Fig. 4a.

When the temperature at which the kinetics is recorded is far from the melting temperature and hence at equilibrium, the concentration of one or several species is very low. The model can therefore be simplified including the equilibrium constant calculated using the temperature-controlled nanospray source. This is the case at 25 °C for $k_{diss-D}$ that was not fitted but replaced by $k_{assoc-D} \times K_D$ in the equations. This allowed to reduce the number of fitted parameters while giving more reliable kinetic constants.

Finally, the natural logarithms of the kinetic constants divided by $T$ are plotted against $1/T$ in an Eyring plot. A linear fit gives access to the enthalpies $\Delta H^\ddagger$ and entropies $\Delta S^\ddagger$ of activation:

$$\ln\frac{k}{T} = -\frac{\Delta H^\ddagger}{RT} + \frac{\Delta S^\ddagger}{R} + \ln\frac{\kappa k_B}{h} \tag{12}$$

Where $k$ is the kinetic constant, $R$ the ideal gas constant (1.987 cal K$^{-1}$ mol$^{-1}$), $T$ the temperature in K, $\kappa$ the transmission coefficient (assumed to be 1), $k_B$ the Boltzmann constant and $h$ Planck's constant.

**Thermographic pictures.** Both the infrared and visible pictures were taken using a FLIR i60 thermographic camera (FLIR Systems AB, Danderyd Sweden). Black tape was placed on the source in order to enhance the blackbody emission coefficient and reduce reflection of light coming from other sources. The emission coefficient used to reconstruct the pictures was 0.95.

**Comsol Multiphysics simulations.** 2D Comsol Multiphysics simulations (COMSOL 5.3 Build 260) were run to estimate the temperature of the liquid inside the capillary, as a function of the position inside the block and of the flow rate by combining fluid dynamics and heat transfer in fluids. The temperatures of the two blocks were fixed by forcing the temperature of one of their sides to 60 and to 25 °C respectively for the back and front blocks. The heat conductivities used were for the blocks (copper) 400 W m$^{-1}$K$^{-1}$, for the capillary (stainless steel) 44.5 W m$^{-1}$ K$^{-1}$ and for the Teflon layer 0.25 W m$^{-1}$ K$^{-1}$. The same flow rates as the ones used experimentally were used in the simulations. The Reynolds numbers for such flow rates in capillaries of 100 µm diameters are low (between 0.06 and 72 depending on the temperature and flow rates) and indicate a laminar flow. A complete report was automatically generated and is provided in Supplementary Table 1. We report the temperature of the liquid at the center of the capillary in Supplementary Fig. 2. In addition, we arbitrarily defined the error on the time axis by the time used to cool to 1K above the temperature of the second block. We also added the time spent in the gap between the blocks to the total error. The absolute errors on the time and the relative errors are presented in Supplementary Table 1.

**Reporting summary.** Further information on research design is available in the Nature Research Reporting Summary linked to this article.

## Data availability
The original data used in this publication are made available in a curated data archive at ETH Zurich (https://www.researchcollection.ethz.ch) under the DOIs 10.3929/ethz-b-000353280 and 10.3929/ethz-b-000385801. The source data underlying Figs. 1a, 2a–d, 6d, h, and 7c and Supplementary Figs. 1a and 5d are provided as a Source Data file. All other data are available from the corresponding author on reasonable request.

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

## Acknowledgements
We thank the Swiss National Science Foundation for the financial support of this research (SNF grant 200020_178765). Christoph André Bärtschi and Christian Marro from the ETH Workshop are acknowledged for the construction of the source. We thank Heinz Benz for building the PID controller and for his help with LabView coding. We also thank Nina B. Hentzen and Prof. Helma Wennemers for providing the collagen model peptide.

## Author contributions
A.M. conceived the source and designed the experiments. A.M. and E.E. optimized the source design and parameters. A.M. performed the experiments and analyzed the data. A.M., M.C., J.K. and R.Z. wrote the paper. A.M., M.C., E.E., J.K. and R.Z. discussed the results and revised the manuscript.

## Competing interests
The authors declare no competing interests.

## Additional information

**Peer Review Information** *Nature Communications* thanks the anonymous reviewers for their contribution to the peer review of this work. Peer reviewer reports are available.

