## [Peer Review File · Nature Communications]

Reviewers' comments:

Reviewer #1 (Remarks to the Author):

The manuscript by Marchand and coworkers describes the use of a novel temperature-jump electrospray source to study the equilibrium and kinetics of biomolecular complex formation and protein folding transitions. The applicability of this source to measure the kinetics of several biomolecular processes is shown, and these data provide a beautiful illustration of the approach. My impression, however, is that the demonstration of model systems probably makes this manuscript better fitted for Analytical chemistry as it seems like a techniques paper to me. That is, I don't believe that the authors have significantly extended what is known about folding and assembly and quite a bit of work in this area, dating to the early days of protein mass spectrometry, has been done by others. I also wonder about the phrasing that the authors have used. In truth, this paper determines some reaction barriers and measures thermodynamic differences between states. This is highly significant and the measurement is beautiful. But, to me this is better called a free energy diagram. Or, in the case of enthalpy it would be a potential energy surface. To claim that a landscape has been completely determined seems like a stretch to me. To do this, the authors would need to perturb the system to see other regions of the surface, which is far more difficult to do experimentally. I don't mean to suggest that this is not important work. It is. It just feels like it fits better in a more specialized journal.

Some additional suggestions...

Results:

Page 5 Paragraph 1: The authors note that the error bars are small for these experiments. However, can the authors clarify if the error is calculated from triplicate analysis or from the calculated error in Table S1? Please clarify this point in the main text.

Page 6 paragraph 1: Does the measured T_m of the DNA G-quadruplex match values from previous experiments in the literature? The authors should provide literature values for each complex or protein studied. Error values should be added to each measured melting temperature as well.

Page 7 Paragraph 2: For DNA G-quadruplex equilibrium experiments, the authors use a temperature ramp of 2 °C/min. This is a rather fast temperature ramp for equilibrium experiments. The authors check to make sure they are at equilibrium by holding the second block at denaturing temperatures and monitoring the distribution at different flow rates (maximum of 32 s). Do the authors expect there to be any other products at longer timescales other than the DNA without the cation (i.e., DNA monomer unfolding or fragmentation)? The reviewers suggest a more inclusive control of holding both of blocks at 75 °C and using the smallest flowrate to ensure equilibrium has been reached at timescales above 32 s.

Page 10 Paragraph 1: The authors attribute the lower average charge of RNase A refolding at short timescales to the dead time of this experiment which is <1 second. However, timescales of 160 ms can be reached with this source. Why can't the folding of RNase A be measured at time points below 1 second?

Page 10 paragraph 3: The authors use Van't Hoff analysis to determine the thermodynamic contributions to the formation of a DNA triplex. Van't Hoff analysis is traditionally used for two-state measurements with a null heat capacity change, yet this system contains more than two states and likely has a heat capacity dependence due to the burying of hydrophilic residues upon triplex formation. The reviewer requests that the authors either describe the weaknesses of Van't Hoff analysis in regards to their experiment, or choose another thermodynamic method that considers the observed multi-state behavior.

Materials and Methods:

Page 20 paragraph 3: The source was kept at a distance of 1 cm away from the MS inlet. The authors claim that the time needed for desolvation is negligible when compared to experimental T-jump timescales. Have the authors examined the effects of changing this distance on the equilibrium or kinetic processes in the systems studied? This could be an additional control to ensure that biomolecular rearrangement or binding is not occurring in the ESI droplets.

Page 21 equations 3 and 4: please define all variables in these equations.

Grammar:

Page 2 paragraph 2: "...the two bounds states dictate..." change bounds to bound.

Page 19 paragraph 2: change copper blocs to copper blocks

Page 21 paragraph 1: "for every systems" should read "for every system"

Page 21 paragraph 1: "average charge stat" should read "average charge state"

Reviewer #2 (Remarks to the Author):

This work describes a major advance for the mass spectrometry-based characterization of biomolecular systems in solution. While previous research groups have developed approaches for recording electrospray mass spectra at different solution temperatures, the current work goes one step further by providing insights into the kinetics of temperature change-induced reactions. I

enthusiastically recommend this work for Nature Communications. My comments (below) are quite minor and will be easy to address.

=== Main Comment ===

The authors have to highlight the novel capabilities of their device more clearly. Simply calling it a T-jump system is an understatement. Most previous T-jump systems are for rapid heating (e.g. Gruebele's work), which severely limits the applicability of those earlier systems. In contrast, the current study can also perform rapid cooling, which is quite rare and has huge potential impact (discussed in Talanta 71 1276-1281 (2007)). Readers will not immediately realize that the device discussed here is for both rapid cooling and a rapid heating. The abstract, introduction and main text should make this more clear, and Figure S1 should be moved to the main text (merge with Fig. 1, perhaps show profiles for both cooling and heating).

=== Minor Comments & Suggestions ===

p. 1: "configurations" should be "conformations"?

p.1 and elsewhere in the text. Be consistent when talking about "energy landscapes" or "potential energy landscapes". The former is probably better.

p.2 and elsewhere in the text: careful when talking about energy vs. free energy vs. Gibbs free energy (e.g. p. 2 "the relative energy levels" should be "... FREE energy levels")

p. 2: provide references when talking about allostery.

p. 2: provide references when talking trapping of inactive states by drugs.

p.2: "multimolecular complexes" should be "receptor-ligand complexes"?

p. 2: “perturb equilibrium, following which the system” check language

p.2: “intra-biomolecular” delete (also on p. 3 and elsewhere in the text. Folding is always “intramolecular”)

On p.2 it should be mentioned that the most common kinetic triggers for protein folding experiments are denaturant-jumps (urea, guanidinium, pH, ...). Others such as pressure-jump are much less common (and not available on commercial stopped-flow instruments).

The T maps of Fig 1 are not very clear. 60/25 seem to refer to the device instead of the legend gradient. Why is (b) crooked?

p.11: “These results confirm that MS is suitable” This statement should be reworded.

p.12: This sentence is unclear: “As previously demonstrated with the collagen triple helix, it is not obligatory to perform jumps from high to low temperatures.”

Figure 4b and elsewhere. It appears that one assumption of the associated analyses is that all species have the same detection (ESI) efficiency. This should be stated explicitly.

p. 15: It is a bit awkward to quote the literature values, without stating the corresponding experimental data (readers don't want to scroll back and forth). Perhaps just state the % agreement here, and move the details to the SI?

p. 16: “typical for mass spectrometry quantification.” A reference is needed.

Related to the two preceding points: The whole Discussion section focuses on rather mundane aspects. Readers looking for profound insights related to energy landscapes etc. (see title!) will be disappointed from this section. Much of the existing text in the Discussion section can be shortened.

Reviewer #3 (Remarks to the Author):

The manuscript describes the application of temperature-jump mass spectrometry to study folding events of biomacromolecules. The area of research is not entirely new. In particular, the Zenobi and the Gabelica groups have published T-based MS folding experiments (reference 8 of the current manuscript), but the manuscript makes nice progress in the quality and accuracy of the method. In particular, faster time scales (down to 160 ms) due to technological advancements are reporting in the current manuscript. The scope of these new methods is nicely demonstrated for a number of different biomolecular systems. I am in principle in favor of publishing the manuscript, once a number of points have been addressed.

Minor points:

1.) p.1, last paragraph: "In chemical reactions involving small molecules, the different states..."

The use of the term "states" is incorrect here, as it is clearly defined by stat. thermodynamics.

2.) p.1, last paragraph: State definition in biomacromolecular folding: states are separated by energetic barriers of interconversion. This definition has to be included.

3.) The relatively broad introduction lacks a discussion of NMR approaches that are clearly competing with mass spectrometry for studying biomolecular folding, e.g. also time-resolved T-jump experiments.

Reviewer #1

[...] To claim that a landscape has been completely determined seems like a stretch to me. To do this, the authors would need to perturb the system to see other regions of the surface, which is far more difficult to do experimentally. [...]

We thank the reviewer for the generally very positive comments.

We agree that the term “landscape” implies that we are probing the energy of the system in multiple regions (e.g. using pressure or denaturant induced-unfolding). With the setup we developed, we are only able to perturb the system by changing its temperature. We therefore only probe one part of the landscape. However, we believe that this aspect is very clear throughout the paper. To make the statement less strong, we removed wording “full energy landscape” throughout.

On the other hand, we believe that this work is of interest for a very broad community because it underscores the utility of native mass spectrometry for the characterization of biomolecular interactions. We believe that the use of native mass spectrometry in such a field will greatly benefit from the use of temperature-controlled sources to characterize both the thermodynamics and kinetics of formation of multimolecular complexes.

Some additional suggestions...

Results:

Page 5 Paragraph 1: The authors note that the error bars are small for these experiments. However, can the authors clarify if the error is calculated from triplicate analysis or from the calculated error in Table S1? Please clarify this point in the main text.

This sentence was indeed confusing. The error bars are determined from the Comsol Multiphysics simulations. We modified the position of the sentence and added the word “computed”, the end of this paragraph should now be more clear:

error is maximum, at 10%; see Table S1). ~~Overall, because these computed error bars in the time dimension are rather small, and so as to not clutter the figures, they are not shown on any of the plots.~~ We note that the time needed for the desolvation process is negligible ($\approx \mu\text{s}$) when compared to the time scales accessible using the T-jump source.³⁹ ~~Overall, because these error bars are rather small, and so as to not clutter the figures, they are not shown on any of the plots.~~

Page 6 paragraph 1: Does the measured T_m of the DNA G-quadruplex match values from previous experiments in the literature? The authors should provide literature values for each complex or protein studied. Error values should be added to each measured melting temperature as well.

It should be noted that the T_m values are highly condition-dependent. The buffer used, its concentration, the concentration of the species... can all affect the observed thermal denaturation temperature. It is therefore difficult to find literature values obtained in the exact same conditions.

However, the comparison of the melting temperatures of the G-quadruplexes obtained using MS and CD was discussed in another paper (Reference 10 of the manuscript: A. Marchand, F. Rosu, R. Zenobi, V. Gabelica, *J. Am. Chem. Soc.* **2018**, 140, 12553–12565.): in general, the T_m values obtained by CD and MS agree within a few degrees. In particular, for the exact sequence and conditions studied here, the T_m value reported using another mass spectrometer is 34 °C, in very good agreement with our value: 36 °C.

In addition, the CD thermal denaturation experiment on the DNA triple helix also shows that, when specific wavelengths are chosen, MS and CD are in very good agreement. (T_m^1 (MS) = 20 °C vs. T_m^1 (CD@269nm) = 21 °C; T_m^2 (MS) = 55 °C vs. T_m^2 (CD@242nm) = 53 °C).

Finally, in the meantime, we published a paper in which we compare the T_m obtained by CD and MS for the denaturation of collagen model triple helices (M. Köhler, A. Marchand, N. B. Hentzen, J. Egli, A. I. Begley, H. Wennemers, R. Zenobi, *Chem. Sci.* **2019**, 68, 42–61.). In this case also, MS and CD show good agreement.

We re-cited reference 10 at the end of page 5 and added a sentence indicating that, in relation to T_m values, MS and CD experiment agree within a few degrees.

Page 5:

assumption of equal detection efficiency⁴⁶ to produce the thermal denaturation curve shown in Figure 2c. The measured thermal denaturation temperature (T_m), defined as the temperature at which half the complexes are formed, is 36 ± 1 °C, in good agreement with the previously reported value.¹⁰

And page 11:

Both thermal denaturation temperatures are very similar to those obtained by CD when monitoring specific wavelengths as previously demonstrated for other DNA complexes¹⁰ and for collagen triple helices⁶⁴ (21 °C at 269 nm and 53 °C at 242 nm for the triplex–duplex and duplex–monomer transitions, respectively; Figure S40S12). When

We also added error bars in the Figures 2, S8, S9, S10 and S11.

Page 7 Paragraph 2: For DNA G-quadruplex equilibrium experiments, the authors use a temperature ramp of 2 °C/min. This is a rather fast temperature ramp for equilibrium experiments. The authors check to make sure they are at equilibrium by holding the second block at denaturing temperatures and monitoring the distribution at different flow rates (maximum of 32 s). Do the authors expect there to be any other products at longer timescales other than the DNA without the cation (i.e., DNA monomer unfolding or fragmentation)? The reviewers suggest a more inclusive control of holding both of blocks at 75 °C and using the smallest flowrate to ensure equilibrium has been reached at timescales above 32 s.

It is indeed important to establish that equilibrium is reached, even at high temperatures, and also to show that additional effects (such as monomer unfolding and fragmentation) do not occur if the DNA is held at high temperature for longer times.

The reviewer suggests one experiment: recording mass spectra at different flow rates using a denaturing temperature in the second block. One part of this

experiment was already performed. In Figure S4, we show the distribution of potassium ions bound at 75 °C using very short residence times. With this experiment, we show that no binding of K⁺ occurs, regardless of the flow rate used.

The reviewer also suggests to include longer residence times to show that no additional effects (such as monomer unfolding and fragmentation) occur. We added a mass spectrum recorded at the lowest flow rate that we could use (in this case, at 75 °C, 1.5 μL/min) to Figure S4. The distribution of K⁺ does not change, though additional TMA adducts appear at low flow rates (as discussed in Figure S5). We note that we cannot use the lowest flow rate (0.5 μL/min) at the highest temperatures because no signal is detected under these conditions.

To ensure that there is no degradation or other effects after **very long times**, we performed an additional experiment. We used the temperature-controlled nanospray source (only one Cu block) and sprayed for 30 minutes at 75 °C. Using this source, the whole solution is maintained at the same temperature for the duration of the experiment (up to 30 minutes). We did not observe any trace of degradation during this experiment.

Concerning the unfolding of the monomer, we recorded the ion mobility of two main charge states (5- and 6-) as a function of time. The mobility of the 0-K⁺ stoichiometry does not change, indicating that its structure does not change. This is in agreement with the fact that, for G-quadruplexes, cation binding is intimately linked to folding and vice versa. This is expected because the CD data (based on the stacking of the bases and therefore the “folding”) and the MS data (based on the number of bound K⁺) always give similar thermal denaturation temperatures. We therefore conclude that equilibrium was indeed reached.

We added two figures to the supplementary materials (Figure S5 and S6) and extended the discussion in the main text:

bound, indicating that complete denaturation is achieved at 75 °C in less than 160 ms (Figure S4). Additionally, we recorded the ion mobility of the 5- and 6- charge states as a function of temperature and time (Figure S5). The mobility of the 0-K⁺ stoichiometry does not change, indicating that the release of K⁺ is concomitant with the unfolding of the G-quadruplex. We also note that no other products were produced after maintaining the DNA at high temperature for longer times (Figure S6).

Page 10 Paragraph 1: The authors attribute the lower average charge of RNase A refolding at short timescales to the dead time of this experiment which is <1 second. However, timescales of 160 ms can be reached with this source. Why can't the folding of RNase A be measured at time points below 1 second?

This comment refers to the sentence in which we mention that the dead time of the experiment is less than 1 second. The sentence was indeed unclear. The dead time of the experiment is 160 ms (which is less than 1 second) and time points below 1 second were indeed used to monitor the refolding of RNase A. We clarified this sentence to read:

Interestingly, at very short time scales, the average charge state is 10+, suggesting that the first step(s) of the refolding of the protein (during which the average charge state changes from 11+ to 10+) occurs in the dead time of the experiment (in this case, ~~less than 1 second~~160 ms). The second part of the kinetics is much slower and well

Page 10 paragraph 3: The authors use Van't Hoff analysis to determine the thermodynamic contributions to the formation of a DNA triplex. Van't Hoff analysis is traditionally used for two-state measurements with a null heat capacity change, yet this system contains more than two states and likely has a heat capacity dependence due to the burying of hydrophilic residues upon triplex formation. The reviewer requests that the authors either describe the weaknesses of Van't Hoff analysis in regards to their experiment, or choose another thermodynamic method that considers the observed multi-state behavior.

We agree that there are limitations to using Van't Hoff and Eyring analysis in the case of a DNA triplex for the reasons stated by the reviewer. We have added a sentence to the paragraph in question that explicitly states this.

the formation of the transition state complexes. The values obtained are shown in Figure 5. We note that the validity of these analyses lies on the assumption that the enthalpies and entropies are only very weakly dependent on the temperature.

Materials and Methods:

Page 20 paragraph 3: The source was kept at a distance of 1 cm away from the MS inlet. The authors claim that the time needed for desolvation is negligible when compared to experimental T-jump timescales. Have the authors examined the effects of changing this distance on the equilibrium or kinetic processes in the systems studied? This could be an additional control to ensure that biomolecular rearrangement or binding is not occurring in the ESI droplets.

We agree that the paper would be made stronger by addressing the issue of emitter position relative to the MS inlet. Normally, the position of the source is adjusted to obtain the highest signal before each experiment. The range of positions that give usable signal is 0.5 to 1.5 cm away from the inlet. On average, a distance of 1 cm is close to the optimum.

We performed the experiment requested by the reviewer. We used a temperature-jump from 75 to 25 °C for 22CTA in 100 mM TMAA and 1 mM KCl. We used a flow rate of 3 μ L/min (residence time = 5.4 s). Under these conditions, refolding is halfway to completion, so that any influence of emitter position should be observable. We added the results as a figure (Figure S20) in the supplementary materials. Notably, there is no observable effect of the distance on the result of the kinetics. This is what we expected because desolvation happens on in the μ s timescale (M. Peschke, U. H. Verkerk, P. Kebarle, J. Am. Soc. Mass Spectrom. 2004, 15, 1424–1434.), whereas the kinetics that we monitor are on the ms timescale.

Mass spectrometry. The source was mounted held in place by a xyz translation stage on a home-made holder in front of a Synapt G2S (Waters, Manchester, UK) at ≈ 1 cm from the MS inlet in order to maximize signal intensity and held in place by a three-way stage. We note that there is no effect of the distance between the outlet of the capillary and the inlet of the MS on the results of the kinetics (Figure S20). This is what we expected because the desolvation occurs on the timescale of the μs range³⁹ whereas the kinetics that we monitor are on the timescale of the ms time scale. The electrospray voltage was set to 3.5 – 3.8 kV using the internal power supply from the instrument. The

Page 21 equations 3 and 4: please define all variables in these equations.

Done.

With M_A and M_B the complementary strands and M_C , the triplex strand, D and T stand for dimer and trimer, respectively, as described in Figure 4a. $k_{\text{assoc-D}}$, $k_{\text{diss-D}}$, $k_{\text{assoc-T}}$ and $k_{\text{diss-T}}$ are the rate constants for the association and the dissociation of the dimer and the association and dissociation of the trimer, respectively, as described in Figure 4a.

Grammar:

Page 2 paragraph 2: "...the two bounds states dictate..." change bounds to bound.

Page 19 paragraph 2: change copper blocs to copper blocks

Page 21 paragraph 1: "for every systems" should read "for every system"

Page 21 paragraph 1: "average charge stat" should read "average charge state"

All corrected.

Reviewer #2

[...] I enthusiastically recommend this work for Nature Communications. My comments (below) are quite minor and will be easy to address.

=== Main Comment ===

The authors have to highlight the novel capabilities of their device more clearly. Simply calling it a T-jump system is an understatement. Most previous T-jump systems are for rapid heating (e.g. Gruebele's work), which severely limits the applicability of those earlier systems. In contrast, the current study can also perform rapid cooling, which is quite rare and has huge potential impact (discussed in Talanta 71 1276-1281 (2007)). Readers will not immediately realize that the device discussed here is for both rapid cooling and a rapid heating. The abstract, introduction and main text should make this more clear, and Figure S1 should be moved to the main text (merge with Fig. 1, perhaps show profiles for both cooling and heating).

We emphasized the fact that the source can be used for jumps up and down in temperature. We added this information in the abstract, the introduction and the discussion of the paper, as suggested by the reviewer. The same notion was already present in the conclusion. We chose not to include the picture of the source in the main text because it would make it very technical.

Abstract:

experiments (0.16 – 32 s) at different temperatures (10 – 90 °C). The setup allows the recording of both folding and unfolding kinetics by using temperature jumps from high to low, and low to high, temperatures. Six biological systems,

End of introduction:

The ability to perform both-temperature jumps from both high to low and low to high temperatures allows one to study thermal unfolding as well as refolding of thermally denatured biomolecular complexes. Studies of biomolecular

Discussion:

reason why temperature jump approaches coupled with NMR have also been developed.^{13,77,78} However, because of technical limitations, for fast temperature changes, only devices allowing jumps from low to high temperature were have been developed thus far, mainly due to technical limitations; this limiting the possible studies to those on unfolding kinetics or renaturation of cold-denatured proteins. Compared to ITC or NMR, relatively low amounts of of rate constants With the source developed here, we allow temperature jumps from high to low are enabled down, which broadens t mass spectrometry gives direct access to individual signals for each of the stoichiometries formed in solution, he range of applications (applicable to the folding of nucleic acids for example). In addition, relatively low

=== Minor Comments & Suggestions ===

p. 1: “configurations” should be “conformations”?
Changed.

p.1 and elsewhere in the text. Be consistent when talking about “energy landscapes” or “potential energy landscapes”. The former is probably better.
The wording was changed to be more consistent. The wording “energy landscapes” was used.

p.2 and elsewhere in the text: careful when talking about energy vs. free energy vs. Gibbs free energy (e.g. p. 2 “the relative energy levels” should be “ ... FREE energy levels”)
The wording was changed to be more consistent. The wordings “free energy” and “Gibbs free energy” were used.

p. 2: provide references when talking about allostery.
p. 2: provide references when talking trapping of inactive states by drugs.
Done. We added two references to review articles related to allosteric inactivation of biomolecules.

p.2: “multimolecular complexes” should be “receptor-ligand complexes”?
We prefer the original wording because it is more general. Receptor-ligand complexes are multimolecular complexes whereas the opposite is not necessarily true. We kept the original wording.

p. 2: “perturb equilibrium, following which the system” check language

Done. The end of the sentence was not necessary and was causing confusion. We have removed it.

~~one to monitor the folding kinetics. Temperature (or, more rarely, pressure) jumps can also be used to quickly perturb equilibrium, following which the system can be monitored. For example, the refolding of cold-denatured proteins can~~

p.2: “intra-biomolecular” delete (also on p. 3 and elsewhere in the text. Folding is always “intramolecular”)

Done. “Intramolecular” was removed when referring to “folding”.

On p.2 it should be mentioned that the most common kinetic triggers for protein folding experiments are denaturant-jumps (urea, guanidinium, pH, ...). Others such as pressure-jump are much less common (and not available on commercial stopped-flow instruments).

We added a paragraph in the introduction.

~~A key element of a kinetics experiment is the trigger used to initiate the reaction. A. The most common way to initiate a reaction is by trigger is the addition of a ligand (i.e., a concentration jump). To study protein folding, denaturants can be added to the solution as triggers to the unfolding kinetics, whereas and re-dilution into the folding buffer allows one to monitor the folding kinetics. Temperature (or, more rarely, pressure) jumps can also be used to quickly perturb equilibrium, following which the system can be monitored. For example, the refolding of cold-denatured proteins can be initiated by a rapid temperature increase, using laser pulses.¹¹⁻¹³ For technical reasons, temperature jump down approaches are more difficult to implement.¹⁴~~

The T maps of Fig 1 are not very clear. 60/25 seem to refer to the device instead of the legend gradient. Why is (b) crooked?

The original picture was slightly rotated. We modified the picture with a slight rotation so that the source is now horizontal. The legend was also modified to make it more readable.

p.11: “These results confirm that MS is suitable” This statement should be reworded. Done.

~~and it is clear that more than two states coexist in solution. These results confirm that MS is a suitable technique to for the study of DNA triplexes, as also previously reported with room temperature experiments.^{65,66}~~

p.12: This sentence is unclear: “As previously demonstrated with the collagen triple helix, it is not obligatory to perform jumps from high to low temperatures.”

We modified the beginning of the paragraph.

~~As previously demonstrated with the collagen triple helix, the source also allows to perform jumps from low to high temperatures. This allowed to it is not obligatory to perform jumps from high to low temperatures. We thus also study examination of~~ the unfolding of these complexes ~~by jumping from low to high temperature.~~ ~~aAs shown in~~

Figure 4b and elsewhere. It appears that one assumption of the associated analyses is that all species have the same detection (ESI) efficiency. This should be stated explicitly.

The reviewer is right. This approximation was stated twice in the materials and methods section but never in the main text. We added a sentence in the main text.

up, the complex unfolds, as indicated by the release of the bound K⁺ cation. The peaks corresponding to the 0 and 1-K⁺ stoichiometries were integrated as a function of temperature and were normalized considering under the assumption of equal detection efficiency⁴⁶ to produce the thermal denaturation curve shown in Figure 2c. The

p. 15: It is a bit awkward to quote the literature values, without stating the corresponding experimental data (readers don't want to scroll back and forth). Perhaps just state the % agreement here, and move the details to the SI?

This is true. We added a table (Table S3) in the supplementary material that compare the values from the literature with our values, emphasizing the differences in the conditions under which the values were measured.

p. 16: "typical for mass spectrometry quantification." A reference is needed.
Done.

Related to the two preceding points: The whole Discussion section focuses on rather mundane aspects. Readers looking for profound insights related to energy landscapes etc. (see title!) will be disappointed from this section. Much of the existing text in the Discussion section can be shortened.

We agree that the discussion is rather long, and the novel aspects may be lost on the reader. We have significantly shortened the discussion throughout so as to make the text more readable.

~~Validation. To validate the rate constants determined using the temperature-jump ESI source, we used different approaches. When possible we compared the values we obtained with literature. In this case, SPR experiments were previously performed to determine the rate constants for the binding of various the ligands we used in this study to Carbonic Anhydrase II and in particular for the ligands we used in this study.⁷¹ Also, the values are in very good agreement for both ligands, in particular for the association constants ($k_{\text{assoc}}^{\text{4-Cbxbz sulfonamide}} = 25800 \text{ M}^{-1} \text{ s}^{-1}$; $k_{\text{diss}}^{\text{4-Cbxbz sulfonamide}} = 0.075 \text{ s}^{-1}$; $k_{\text{assoc}}^{\text{Bzsulfonamide}} = 9180 \text{ M}^{-1} \text{ s}^{-1}$; $k_{\text{diss}}^{\text{Bzsulfonamide}} = 0.01025 \text{ s}^{-1}$ within 50%).⁷¹ The slight differences could come from the fact that the ionic strength that was not identical for the two studies (10 vs. 50 mM NH₄OAc). The intramolecular folding of Ribonuclease A was studied using NMR temperature jumps and the reported rate constant is $k_{\text{fold}} = 2.8 \times 10^{-2} \text{ s}^{-1}$ at 29 °C,¹¹ also in very good agreement with our value.¹³ The values are in very good agreement for both systems (See Table S3 for a comparison of the values).⁷¹~~

We also moved the discussion subtitle. We believe that the discussion on the folding landscape was previously misplaced in the results section. The discussion is now made of the following parts: Discussion on the folding landscape, validation, reproducibility, comparison with other techniques, limitations.

Reviewer #3

[...] I am in principle in favor of publishing the manuscript, once a number of points have been addressed.

Minor points:

1.) p.1, last paragraph: “In chemical reactions involving small molecules, the different states...”

The use of the term “states” is incorrect here, as it is clearly defined by stat. thermodynamics.

The wording “state” / “states” was modified (see next comment).

2.) p.1, last paragraph: State definition in biomacromolecular folding: states are separated by energetic barriers of interconversion. This definition has to be included. We agree that in this paragraph, we improperly used the word “state”. We modified the paragraph as follows:

Studying ~~potential~~-energy landscapes of biomolecules and biomolecular complexes is not straightforward. In chemical reactions involving small molecules, the different ~~states-species~~ are structurally and energetically well-defined because of the formation of high-energy covalent bonds; in contrast, the folding of a large biomolecule is driven by multiple noncovalent interactions. In this case, ~~different-one observable~~ ~~states-of-a-biomolecule-should~~ ~~therefore-be-seen-as~~ ~~could-correspond-to-ensembles-may-be-an-ensemble-of~~ ~~several-microstates~~ sharing similar thermodynamic and kinetic properties.⁴ The picture can be even more complicated when binding partners are

3.) The relatively broad introduction lacks a discussion of NMR approaches that are clearly competing with mass spectrometry for studying biomolecular folding, e.g. also time-resolved T-jump experiments.

This is true. We chose to have a quite broad introduction to attract the reader. We decided not to include any detailed introduction on specific techniques in the introduction. On the other hand, we have a paragraph in which we compare TJump mass spectrometry to more conventional techniques used to study kinetics.

The reviewer is, however, correct to note that we did not dedicate enough space to NMR. We have extended the discussion on NMR in the discussion section.

~~To study kinetics of biomolecular assemblies, one of the challenges is to resolve different ensembles. This is the reason why temperature jump approaches coupled with NMR have also been developed.^{13,77,78} However, because of technical limitations, for fast temperature changes, only devices allowing jumps from low to high temperature were have been developed thus far, mainly due to technical limitations; this limiting the possible studies to those on unfolding kinetics or renaturation of cold-denatured proteins. Compared to ITC or NMR, relatively low amounts of samples (150 μ L of 10 μ M concentration or 1.5×10^{-9} moles per kinetic run) are required for MS and, unlike in SPR, there is no need to immobilize one of the binding partner to a surface.~~

~~When compared to other devices that can be coupled to MS (mixing tee/stopped flow) to study folding kinetics, the main advantage of our source is the ability to record kinetics at different temperatures. Temperature control has also been implemented for mixing approaches but only very slow kinetics could be monitored.²⁸ Using mixing devices, organic solvents are often used to induce the folding or unfolding and, even if diluted, the co-solvent will always be seen as a contaminant. The mixing itself is also sometimes not perfect. In addition to giving access to a large range of rate constants, With the source developed here, we allow temperature jumps from high to low are enabled down, which broadens mass spectrometry gives direct access to individual signals for each of the stoichiometries formed in solution, the range of applications (applicable to the folding of nucleic acids for example). In addition, relatively low amounts of samples (150 μ L of 10 μ M concentration or 1.5×10^{-9} moles per kinetic run) are required.~~

Additional changes:

- Based on the Editorial policy checklist: explanation of the error bars of Figure 5.
- Addition of an author (previously acknowledged): Jérôme Kaeslin. His help throughout the development of the project was invaluable.
- Minor English polishing
- 8 references were added:

On GPCR allosteric inhibition:

6. Wootten, D., Christopoulos, A. & Sexton, P. M. Emerging paradigms in GPCR allostery: implications for drug discovery. *Nat. Rev. Drug Discov.* 12, 630–644 (2013).

7. Digby, G. J., Conn, P. J. & Lindsley, C. W. Orthosteric- and allosteric-induced ligand-directed trafficking at GPCRs. *Curr. Opin. Drug Discov. Devel.* 13, 587–94 (2010).

On Temperature-jump triggered by rapid cooling:

14. Boys, B. L. & Konermann, L. A temperature-jump stopped-flow system for monitoring chemical kinetics triggered by rapid cooling. *Talanta* 71, 1276–1281 (2007).

On the MS detection efficiency of biomolecular complexes:

46. Gabelica, V., Rosu, F. & De Pauw, E. A simple method to determine electrospray response factors of noncovalent complexes. *Anal. Chem.* 81, 6708–15 (2009).

On temperature-controlled MS:

64. Köhler, M. et al. Temperature-controlled electrospray ionization mass spectrometry as a tool to study collagen homo- and heterotrimers. *Chem. Sci.* 68, 42–61 (2019).

On errors of calculating affinity constants using MS:

72. Rosu, F., De Pauw, E. & Gabelica, V. Electrospray mass spectrometry to study drug-nucleic acids interactions. *Biochimie* 90, 1074–1087 (2008).

On temperature-jump NMR:

77. Gal, M., Zibzener, K. & Frydman, L. A capacitively coupled temperature-jump arrangement for high-resolution biomolecular NMR. *Magn. Reson. Chem.* 48, 842–847 (2010).

78. Kawakami, M. & Akasaka, K. Microwave temperature-jump nuclear magnetic resonance system for aqueous solutions. *Rev. Sci. Instrum.* 69, 3365–3369 (1998).

REVIEWERS' COMMENTS:

Reviewer #2 (Remarks to the Author):

The authors have addressed all of the Reviewer Comments. The manuscript can be accepted as is.

Reviewer #3 (Remarks to the Author):

I recommend publication of the manuscript in its current form.